# An Identity Authentication Method of a MIoT Device Based on Radio Frequency (RF) Fingerprint Technology

**DOI:** 10.3390/s20041213

**Published:** 2020-02-22

**Authors:** Qiao Tian, Yun Lin, Xinghao Guo, Jin Wang, Osama AlFarraj, Amr Tolba

**Affiliations:** 1College of Computer Science and Technology, Harbin Engineering University, Harbin 150001, China; tianqiao@hrbeu.edu.cn; 2College of Information and Communication Engineering, Harbin Engineering University, Harbin 150001, China; linyun@hrbeu.edu.cn (Y.L.); s317080023@hrbeu.edu.cn (X.G.); 3Hunan Provincial Key Laboratory of Intelligent Processing of Big Data on Transportation, School of Computer & Communication Engineering, Changsha University of Science & Technology, Changsha 410004, China; 4School of Information Science and Engineering, Fujian University of Technology, Fujian 350118, China; 5Computer Science Department, Community College, King Saud University, Riyadh 11437, Saudi Arabia; oalfarraj@ksu.edu.sa (O.A.); atolba@ksu.edu.sa (A.T.); 6Mathematics and Computer Science Department, Faculty of Science, Menoufia University, Shebin-El-kom 32511, Egypt

**Keywords:** RF fingerprint, identity authentication, mobile Internet of Things, feature exaction

## Abstract

With the continuous development of science and engineering technology, our society has entered the era of the mobile Internet of Things (MIoT). MIoT refers to the combination of advanced manufacturing technologies with the Internet of Things (IoT) to create a flexible digital manufacturing ecosystem. The wireless communication technology in the Internet of Things is a bridge between mobile devices. Therefore, the introduction of machine learning (ML) algorithms into MIoT wireless communication has become a research direction of concern. However, the traditional key-based wireless communication method demonstrates security problems and cannot meet the security requirements of the MIoT. Based on the research on the communication of the physical layer and the support vector data description (SVDD) algorithm, this paper establishes a radio frequency fingerprint (RFF or RF fingerprint) authentication model for a communication device. The communication device in the MIoT is accurately and efficiently identified by extracting the radio frequency fingerprint of the communication signal. In the simulation experiment, this paper introduces the neighborhood component analysis (NCA) method and the SVDD method to establish a communication device authentication model. At a signal-to-noise ratio (SNR) of 15 dB, the authentic devices authentication success rate (ASR) and the rogue devices detection success rate (RSR) are both 90%.

## 1. Introduction

The Internet of Things is the “internet connected to everything”, which is an extended and expanded network based on the Internet. It combines various information sensor devices with the internet and forms a huge network to realize the interconnection of people, machines and things at any time and any place [1]. In the 21st century, the internet, new energy, new materials, and biotechnology are forming huge industrial capacities and markets very quickly, and this will elevate the entire social production system to a new level and promote a new industrial revolution. In 1999, American Auto-ID first put forward the concept of the “Internet of Things”, which is mainly based on item coding, RFID technology, and the internet [2]. Some experts and scholars have put forward the concept ofthe mobile Internet of Things (MIoT) on the basis of integrating mobile devices and Internet of Things. In the MIoT’s communication mode, wireless communication is very important.

In the field of MIoT research, many scholars have focused on improving the transmission performance of the network [3,4,5], and this article focuses on the security of the network. There are many ways to improve network security, including radio frequency (RF) fingerprinting technology. A RF fingerprint refers to the difference of the transmitters due to the production, processing, and debugging. The signal received from the transmitters can be used to extract the difference and to realize the individual identification of wireless devices. The process of extracting signal differences is called RF fingerprint extraction [6]. RF fingerprint authentication technology aims at distinguishing authorized transmitters of multifarious users based on the unique feature from their radio frequency signals at the physical layer [7]. The core is that only authorized users can be allowed to intervene in the network, which improves the network security to a certain extent. RF fingerprint techniques have been used in many systems such as intrusion detection systems, radar systems, satellite communication systems, Internet of Things systems, and network security in 4G and 5G networks.

Artificial intelligence is a concept that began to be widely discussed in 1956. With the continuous development of artificial intelligence technology, machine learning and deep learning gradually occupy all aspects of people’s lives. In the field of communication, the development of artificial intelligence is also very rapid, including wireless sensor networks [8,9,10,11], network resource allocation [12], modulation signal recognition [13,14,15] communication equipment individual recognition [16], abnormal information identification and detection [17,18], and others. Many scholars have conducted studies in these aspects. This paper studies the use of artificial intelligence algorithms to solve identity authentication problems in the MIoT. Generally, the process of RFF technology contains 4 primary procedures: signal processing and acquisition, features extraction from the gained signals, features matching with dataset of fingerprint reference, and allocating the best matching collection for those features. In the final phase, if there is no appropriate class, signals under check ought to see as malicious signals of illegal transmitters and banned from network access.

The main purpose of the research in this paper is to establish the radio frequency authentication model and the radio frequency fingerprint database through the research on radio frequency fingerprint technology, to effectively distinguish between authentic devices and rogue devices in wireless communication. Based on our recent research on the application of the RF fingerprint authentication model to MIoT, we arrange the structure of the paper as follows: in Section 2, we systematically summarize and classify the research on the RF fingerprint authentication model and feature extraction algorithm from the past few years. Then, in Section 3, we list the principles and details of some methods used in this paper. In Section 4, we conduct simulation experiments on mobile device authentication models based on measured data and evaluate their performance. Finally, we summarize the whole thesis in Section 5.

## 2. Related Work

Based on the research of RF fingerprint related fields, this paper focuses on two important aspects: (1) the RF fingerprint feature extraction of wireless devices; (2) RF fingerprint authentication. The radio frequency fingerprint feature extraction is to perform feature transformation and feature extraction on the identifiable part of the communication signal, and convert the communication signal into a radio frequency fingerprint feature that can be identified.

### 2.1. RF Fingerprint Feature Extraction

The following is a summary of the recent work on RF fingerprint feature extraction methods for wireless devices:

Existing methods for the RF fingerprint extraction contain numerical feature extraction established on signal constraints, based on signal transform domain, based on nonlinear characteristic of transmitter and based on the methods used in image processing. The feature extraction method based on the numerical features of the signal refers to finding a linear or nonlinear conversion to reveal the essential structure of the preprocessed signal, and project the imaginative signal into the unique feature space, this can decrease the dimension of the source signal, and reduce the problem of over fitting with classifiers. The common signal factors consist of time domain parameters, high order spectral parameters, domain frequency parameters, high order moments parameters, and so on. In collected works [19,20], researchers identify the transient signals amplitude by using IEEE 802.11 and Bluetooth devices. The detection rate average can extent to 95%, however the complexity time is relatively high. In [21], the I/Q modulation domain imbalance is used as a RF fingerprint. The idea is constructed on the authentication of the 2 parameters assumption test and the probability ratio test.

The final simulation result proved the correctness of the idea. In the year of 2015, the researcher of [22] take out the information phase of the baseband signal by straining the signal with alike model from the similar manufacturer, and then used it as the RF fingerprint. The results display that information phase be able to use to categorize devices, but the classification performance will change owing to the difference of the channel distance. The accuracy average of classification in distance with short range is about 99.6%, however the accuracy average of classification reduced to the level 81.9% after the channel distance becomes elongated. The entropy represents the internal chaos degree in a system. Whenever the chaos increased, the entropy gets a higher value. The characteristics of entropy are frequently used features in RF fingerprint. For instance, the authors of [23] proposed recognition method of a new fingerprint established on multi-dimensional permutation entropy. According to this experiment, the transceiver distance is fixed to 10 meters, thus the signal can be propagated in the channel of short-wave line of sight. Experimental results show that the method is efficient.

On behalf of weak dissimilarities among like devices, transmitter hardware internal noise and nonlinearities be able to produce spurious modules at the signal receiving. Most of all signal modules are non-Gaussian and non-stable, consequently numerical analysis approaches for frequency domain and time domain parameters might no longer be appropriate. Therefore, scholars gradually use methods of signal processing for analyzing signals, convert them into assured transform domains, and then process and analyze them. Including analysis of wavelet, analysis of time frequency, fractal features, transformations of empirical mode decomposition (EMD), and the transformations of intrinsic time decomposition (ITD), etc. In the late of 1998, Huang et al. [24] introduced a method of data analysis constructed on EMD, which be able for generating a set of intrinsic mode functions(IMF). The transformation of Hilbert is able to use for deriving instantaneous frequency and local energy from IMF for obtaining a complete distribution of the energy frequency time. But, the authors in [25] point out that EMD has some shortcomings. For instance, the process to obtain the IMF in EMD is ineffective and there are serious boundary effects in it. The more importantly though is that the process of EMD produces new modules which do not present in the source signal. In the year 2012, Klein [26] utilized a dual-tree complex wavelet transform (DT-CWT) feature extracted from a non-transient preamble response of an OFDM established on 802.11a signal for identifying four devices with same model by Cisco with dissimilar serial numbers. The accuracy of classification be able to reach 80% when the SNR is lower than 20 dB.

Like many technologies, RF fingerprinting technology is derived from military technology, which can be traced back to the identification of enemies and radar in World War II. The main enemy judgment is made by directly comparing the waveform of the received signal with the waveform map that has been registered by our radar. However, with the increase of equipment and the improvement of production processes, it is impractical to directly compare the signal waveforms. As early as 1995, Choe and Toonstra began to study the characteristics of the extracted device communication signals to detect illegally operating VHF FM transmitters [27,28]. Subsequently, a large number of RF fingerprinting technologies began to be researched, and various RF fingerprint extraction and authentication methods emerged.

In 2018, Peng et al. designed a hybrid and adaptive classification scheme adjusting to the environment conditions, and carried out extensive experiments to evaluate the performance [29]. They constructed a testbed using a universal software radio peripheral platform as the receiver and 54 ZigBee nodes as the candidate devices to be classified, which is the most ZigBee devices ever tested. The classification error rate is as low as 0.048 in the LOS scenario, and 0.1105, even when a different receiver is used for classification, 18 months after the training. In 2019, Wang et al. selected different characteristics of RF fingerprints and compareed the identification accuracy of Zigbee devices with five classification algorithms [30]. The experimental research shows that the highest identification accuracy reached approximately 100% by using multi-features of frequency offset, IQ offset, and circle offset based on the neural network algorithm under a high SNR.

### 2.2. RF Fingerprint Authentication

In the research of RF fingerprint authentication, device ID verification is an important research content [31,32,33]. As a device identity detection for a declared ID number, it can be applied to RF fingerprint authentication. This one-to-one verification is computationally intensive and is ideal for lightweight authentication devices. Device ID verification can correctly provide network access to authorized users, as well as network access requests from malicious devices. Therefore, this research has received extensive attention from more and more scholars.

Authorization of network devices has been a serious problem when it comes to access to infrastructure. Dubendorfer, based on the research of Zigbee “hacker” tools, found that the existing anti-attack methods have considerable security risks [31]. First, he collected the transient signals of Zigbee devices and used the RF-DNA fingerprints of seven authorized devices to perform multiple discriminant analysis (MDA) training and identifies these devices. The device’s network access rights are then determined by verifying the device’s claimed identity to filter the rogue device. For authorized devices, a test statistic based on the hypothetical multivariate Gaussian (MVG) likelihood values is used. At SNR = 5 dB, the detection rate of the rogue device is 85%; at SNR = 10 dB, the detection rate exceeds 90%.

In 2012, Cobb proposed that device identification and verification was done by passively monitoring and utilizing the inherent characteristics of IC unintentional RF transmissions, without any modifications to the device being analyzed [32]. He used multiple discriminant analysis to train the recognition system and reduce the data dimension, and used a linear Bayesian classifier for device ID verification. Then, by comparing the Bayesian posterior probability with a specific threshold, it is verified whether the identity of the device is consistent with the classification result. The identification and verification simulation of this study consisted of 40 devices of the same model. At 10 dB, the average verification rate reached 99% and the test error rate was less than 0.05%.

With the popularity of Zigbee equipment in home automation, transportation and industrial control systems, its safety has also received more and more attention. In 2015, Patel analyzed the past Fisher-based multi-discriminant analysis and maximum likelihood (MDA-ML) classification and verification process in detail, and pointed out its problem: When the distribution of RFF does not meet the Gaussian normal condition, the performance of MDA-ML will be reduced [33].

This paper proposes introducing nonparametric random forest and multi-class AdaBoost integrated classifiers into classification and authentication process of devices. In performance testing, this paper used four authorized ZigBee devices for classifier training. Nine rogue devices that have not been seen before have been introduced to measure the performance of the classifier. At 10 dB, the error classification probability is less than 10%.

Unauthorized network access and fraud attacks have been the main research task of information technology security in wireless network communications. In 2015, Reising attempted to solve this problem using RF fingerprint technology to enhance WAP security [34]. This paper proposes the detection of malicious devices posing as authorized devices by means of dimensionality reduction analysis (DRA) and device ID authentication. Moreover, in recent years, research on the application of RF fingerprint technology in other fields has begun to emerge [35,36,37], which has enriched the application range of RF fingerprint technology and removed some obstacles for subsequent engineering applications.

In reference [38], a lightweight terminal identity authentication scheme is proposed. The authors use a method based on radio frequency fingerprint identification (RFFID) to authenticate terminal devices with limited computing capabilities in the Internet of Things. In real scenarios, the wavelet features used by this method have a high recognition rate, and prove the effectiveness of this authentication scheme.

ZigBee devices are widely used in the field of IoT. However, more and more scholars pay attention to the authentication security of ZigBee AD hoc network’s decentralized architecture. Reference [39] applied the RF-Distinct Native Attribute (RF-DNA) method to device authentication at low SNR. Multipath channel and interference from other devices are considered in the research. The authors used nonparametric random forest and multi-class AdaBoost to authenticate ZigBee devices.

Previous work has focused on using RFF technology to authenticate mobile devices [29,30,31,32,33,34,35]. This authentication method requires the device to be authenticated to provide a claimed identity in advance. The authentication model will compare the RFF of the device with the RFF of the device that claims the identity. This comparison results in a judgment as to whether the device to be authenticated is legal. However, the method in this paper does not require the device to be authenticated to provide a claimed identity. It only needs to obtain the RFF of the device to be authenticated, and the authentication model in this paper can handle the legality of the identity of the authentication device to make a decision. This enables the method in this paper to make a decision without other prior information, which further enhances the security of the RFF authentication algorithm.

## 3. Method

### 3.1. NCA Feature Selection

Neighborhood component analysis (NCA) is a non-parametric method for selecting features with the goal of maximizing prediction accuracy of regression and classification algorithms. The important of this method is to catch positive definite matrix *H* connected to the spatial transformation matrix, which can be gotten by characterizing differentiable objective function of *H* also, by methods for iterative strategies. One of the advantages of this algorithm is that the categories number K can be well-defined by a function f (scalar constants determination). Thus, the problem of model selection can be solved by this algorithm. To describe the transformation matrix *H*, initially we explain the objective function f that illustrates the accuracy of classification in the target transformation matrix, and attempt to determine that $H^{*}$ make best use of this objective function.
(1)$$H^{*}=\arg\max_{H}f\left(H\right)$$

Once classifying a single data point, we want to consider the k-nearest neighbors (k-NN) controlled by a given separation metric, and acquire the sample class according to the labeled category of the k neighbors. In the target transformation space, we don’t utilize the method of left-sort to find k nearest neighbors for each sample point, however the entire data set is considered as a random nearest neighbor in the target space. Moreover, squared Euclidean distance function has been utilized to describe the distance among a data point and additional data in the target transformation space. The function can be defined as:(2)$$p{}_{ij}=\left\{\begin{array}{c} \frac{e{}^{-∥Hx{}_{i}-Hx{}_{j}∥{}^{2}}}{\sum_{k}e{}^{-∥Hx{}_{i}-Hx{}_{j}∥{}^{2}}},\mathrm{if}\;j\neq i\\ 0,\mathrm{if}\;j=i \end{array}\right.$$

The classification accuracy of the input point *i* is the classification accuracy of the nearest neighbor set $C_{i}$ adjacent to it: $p_{i}=\sum_{j}^{n}p_{ij}$, where $p_{ij}$ denotes the probability that *j* is the nearest neighbor of *i*. The objective function, defined by the global data set as the nearest neighbor classification method of random nearest neighbors, is defined as follows:(3)$$f\left(H\right)=\sum_{i}\sum_{j\in C_{i}}p_{ij}=\sum_{i}p_{i}$$

The objective function can be better chosen as:(4)$$\frac{\partial f}{\partial H}=-2H\sum_{i}\sum_{C_{i}}p_{ij}\left(x_{ij}{x_{ij}}^{T}-\sum_{k}p_{ik}x_{ik}{x_{ik}}^{T}\right)$$

The continuous gradient descent algorithm is used here.

### 3.2. Support Vector Data Description

SVDD (support vector data description) is established based on statistical learning theory, inherits its advantages and develops continuously, and has a very complete theoretical foundation and basis. In 2004, Tax and Duin conducted further expansion and more complete research on SVDD, and obtained SVDD without negative samples and with negative samples [40]. Wei et al. introduced the algorithm of SVDD with Markov distance as a measure to replace the traditional Euclidean distance and establish the hyperellipsoid to solve the problem that the hypersphere could not cover some training samples well in the process of fault diagnosis [41]. Zhang et al. proposed a fault diagnosis framework of analog circuits based on a single classifier, and introduced “or” combined results into the test samples to solve the fault in the overlapping region of test samples [42]. In recent years, some research on SVDD algorithm has been published [43,44], and further reasonable improvements have been made to the SVDD algorithm.

The basic idea of SVDD is: Given a data set $X=\{x_{1},x_{2},\cdots x_{n}\}$ containing *n* sample points, the goal of SVDD algorithm is to find a minimum circle with a as the center and *R* as the radius; the circle can contain all or as many sample points in *X* as possible. The point where the distance from the training data to the center of the circle is equal to the radius is called the support vector, so the optimization problem can be described as:(5)$${\begin{array}{cc} s.t. & \left(x_{i}-a\right)\left(x_{i}-a\right) \end{array}}^{T}\leq R^{2}+\xi_{i}$$
where a is the center of the circle; R is the radius; $\xi_{i}\geq 0$ is the slack variable; and C>0 is the penalty factor, which is used to achieve a comprehensive adjustment between the size of the circle and the number of samples included. The geometric model of SVDD is shown in Figure 1. The black dots in the graph are the data samples in the set *X*.

The above optimization problem can be solved by the Lagrange multiplier method to construct the Lagrange equation:(6)$$\begin{array}{c} L\left(R,a,\alpha_{i},\xi_{i}\right)=R^{2}+C\sum_{i=1}^{n}\xi_{i}-\\ \sum_{i=1}^{n}\alpha_{i}\left(R^{2}+\xi_{i}-\left(x^{2}-2ax_{i}+a^{2}\right)\right)-\sum_{i=1}^{n}\gamma_{i}\xi_{i} \end{array}$$

From the above equation:(7)$$\begin{array}{c} W=\min_{\alpha }\;\sum_{i=1}^{n}\alpha_{i}\left(x_{i}\cdot x_{i}\right)-\sum_{i=1}^{n}\sum_{j=1}^{n}\alpha_{i}\alpha_{j}\left(x_{i}\cdot x_{j}\right)\\ \begin{array}{cc} s.t. & \sum_{i=1}^{n}\alpha_{i}=1 \end{array}\\ 0\leq \alpha_{i}\leq C\left(\forall i=1,2,\cdots ,n\right) \end{array}$$

When the input space is non-circular, the kernel function is introduced to improve the applicability of the algorithm. We find a suitable mapping φ to map the input sample $x_{i}$ to a high-dimensional feature space $\varphi (x_{i})$, and find a hypersphere in the high-dimensional space to surround as many points in the input space as possible. Therefore, the inner product $\left(x_{i}\cdot x_{j}\right)$ in the above equation can be replaced by the kernel function $k(x_{i}\cdot x_{j})$, and the Gaussian kernel function is selected in this paper. At this point, Equation (Equation 8) can be converted into a Lagrange dual problem:(8)$$\begin{array}{c} W=\max_{\alpha }\;\sum_{i=1}^{n}\sum_{j=1}^{n}\alpha_{i}\alpha_{j}k\left(x_{i}\cdot x_{j}\right)-1\\ \begin{array}{cc} s.t. & \sum_{i=1}^{n}\alpha_{i}=1 \end{array}\\ 0\leq \alpha_{i}\leq C\left(\forall i=1,2,\cdots ,n\right) \end{array}$$

Equation (Equation 9) is a typical quadratic optimization problem, and its decision function is:(9)$$f\left(x_{i}\right)=\left({∥\varphi (x_{i})-a∥}^{2}-R^{2}\right)$$

According to the above equation, when $f(x_{i})=-1$,$x_{i}$ is classified as normal data point. When $f(x_{i})=1$, $x_{i}$ are classified as outlier data points.

### 3.3. Whale Swarm Optimization Algorithm

As the parameters of penalty parameter C and kernel parameter g play a very critical role in the performance of the SVDD model, we need to find suitable parameters to make the performance of the SVDD model better. This paper uses the whale optimization algorithm to optimize the parameters C and g in the SVDD model. Mirjalili proposed the whale swarm optimization algorithm (WOA) in 2016 [45]. This algorithm was inspired by humpback whales using a “spiral bubble net” strategy for hunting. The position of each humpback whale represents a feasible solution. The algorithm has the advantages of less adjustment parameters, simple operation, and strong local optimal ability.

WOA is a mathematical model of humpback whales based on three behaviors: surrounding prey, hunting behavior, and random hunting behavior.

(1) Surround the prey. According to the optimized model established by Mirjalili et al., after finding prey, humpback whales can quickly surround the prey and constantly update the position. The mathematical expression of the position update is
(10)$$\vec{D}=\left|\vec{C}{\vec{X}}^{*}\left(t\right)-\vec{X}\left(t\right)\right|$$
(11)$$\vec{X}\left(t+1\right)={\vec{X}}^{*}\left(t\right)-\vec{A}\vec{D}$$
where *t* is the current iteration number; ${\vec{X}}^{*}$ is the best position space for the current whale population; $\vec{X}$ is the location space of individuals; $\vec{A}$, $\vec{C}$ are coefficients, and $\vec{D}$ is the distance between the current individual and the target.

The calculation method of $\vec{A}$ and $\vec{C}$ is
(12)$$\vec{A}=2\vec{a}\vec{r}-\vec{a}$$
(13)$$\vec{C}=2\vec{r}$$
(14)$$\vec{a}=2-2j/M$$
where *a* is the vector that decreases linearly from 2 to 0 during iteration; *R* is a random number between 0 and 1; and *M* is the maximum number of iterations.

(2) Hunting behavior. Humpback whales hunt in a spiral motion.
(15)$$\vec{D^{\prime }}=\left|{\vec{X}}^{*}\left(t\right)-\vec{X}\left(t\right)\right|$$
(16)$$\vec{X}\left(t+l\right)=\vec{D^{\prime }}e^{bl}\left(\cos2\pi l\right)+{\vec{X}}^{*}\left(t\right)$$
where *b* is a constant used to define the helical shape; *l* is a random number between −1 and 1, and $\vec{D^{\prime }}$ is the distance between the best individual and the target.

(3) Search for prey. The model of the whale group is
(17)$$\vec{D}=\left|\vec{C}{\vec{X}}_{\mathrm{rand}}-\vec{X}\left(t\right)\right|$$
(18)$$\vec{X}\left(t+1\right)={\vec{X}}_{\mathrm{rand}}-\vec{A}\vec{D}$$
where ${\vec{X}}_{\mathrm{rand}}$ is the position vector of the randomly selected whale population.

Based on the SVDD model and WOA parameter optimization algorithm, this paper proposes an integrated SVDD model. The specific operation steps of this model are as follows: During the training phase of integrated SVDD model, users need to store the samples of the training set separately according to categories, and convert the original signals into RFF features through RFF feature extraction and a selection module. The next step is to input the extracted RFF into the SVDD single-category certification model, which also requires separate training according to the category, that is, the RFF characteristics of each type of sample are trained into a SVDD single-category certification model. The training portion of the integrated SVDD model is completed by training multiple SVDD single-category certification models and combining them.

During the integrated SVDD model test phase, we used the same RFF feature extraction and selection method as the training set to convert the communication signals of the test equipment into RFF features. The second step requires the help of the nearest neighbor finder. By inputting the RFF of the test device into the nearest neighbor finder, we obtain the RFF cluster of A certain type of training sample closest to the RFF of the device in the training set (assuming such a training sample is category A). Finally, we input the RFF into the SVDD single-category authentication model corresponding to category A. As the output of the SVDD single-category authentication model is binary, we can determine whether the test device belongs to category A based on the output. As category A is the training sample, the devices in the training set in the RFF authentication model are all registered legal devices. Therefore, if the test device has passed the identity authentication of the SVDD single-category authentication model corresponding to category A, it is legitimate device. Otherwise, it is a rogue device.

## 4. Experiment

### 4.1. Experimental Environment and Experimental Devices

The content of the previous chapter introduced some basic theories of RFF authentication systems. Our actual MIoT operation usually involves the following scenario: When a mobile device or user wants to access the MIoT to complete practical tasks, it needs to complete the authentication operation first. The authentication system needs to verify whether this device is a registered device in the background database, complete normal authentication operation for the registered device, and terminate its authentication operation for the unregistered rogue device.

As the methods and strategies proposed in this paper are based on the physical layer, it means that we can only obtain the subtle fingerprint characteristics from the received signal of the device to determine its identity. For the registered device in the background database, we will record its signal fingerprint characteristics by receiving the signal of the device several times. For unregistered rogue devices, we do not know their signal characteristics in advance. The purpose of this section of the experiment is to establish a complete set of registration and authentication systems by modeling the signals of real mobile devices. We perform normal authentication operations on registered authentic devices and prevent unregistered rogue devices from logging in through the authentication system.

In the experiment of this section, the authors prepared 10 wireless devices of the same model as experimental devices and numbered them uniformly (#1–#10). The wireless devices’ model is the Motorola walkie-talkie A12. The experimental setup is shown in Figure 2. We use cables to connect the Agilent oscilloscope to the wireless device to collect its transient signal. Then add Gaussian white noise manually. In order to efficiently obtain the subtle differences between the signals of different experimental device, we extracted the instantaneous amplitude envelope of the signal using a Hilbert transform and performed the 50:1 sampling process. This method can reduce the computation burden of the authentication system without affecting the experimental results.

The authentication system is divided into two parts: the registration part and the authentication part. During the experiment, the authors divide the devices into two groups, one is the authentic devices (number of devices is 8), and the other is the rogue devices (number of devices is 2). Among them, the authentic devices are devices registered with the server, and the rogue devices are devices that are not registered with the server. During the registration phase, the oscilloscope receives the signals of the authentic devices multiple times, performs RF fingerprint feature extraction on the signals, and stores the extracted RF fingerprint features according to the device categories. We then use the RF fingerprint training of the authentic devices to train the authentication system model. In the authentication phase, we not only use authentic devices for device login and authentication, but also use unregistered rogue devices to try to log in to the system.

The above content introduces the experimental equipment and experimental procedures. Next, we need to know the performance indicators of the authentication model. There are two main performance indicators: authentic devices authentication success rate (ASR) and rogue devices detection success rate (RSR). ASR refers to the rate that the authentication system recognizes it as the correct legal identity when the authentic devices log in to the system. For example, when the #1 device authenticates, the system recognizes it as the #1 device. RSR refers to the rate that the system can detect it as an unregistered device when rogue devices log in to the system. The results of the next experiment are also based on the performance of these two indicators.

### 4.2. Simulation Analysis

In this paper, the signals of 10 wireless devices were collected as research samples. First, we intercepted the original signal to obtain the power-on transient signals of 10 wireless devices. Regarding the interception of transient signals, this paper uses the findchangepts function in Matlab 2019a to complete the purpose of change point detection. Figure 3 shows the power-on transient signals of four of these devices. We can see that the difference between the transient signals of the four devices is not obvious. Then, we extracted the amplitude envelope of the transient signal through the Hilbert transform. The principal component analysis (PCA) method was used to reduce the dimension of amplitude envelope to obtain the signal features. Finally, the NCA feature selection method was used to screen the signal features obtained in the previous step to obtain the RFF features. In this paper, the NCA feature selection algorithm was used to reduce the dimension of amplitude envelopes. Features with weight w greater than 1 in the NCA algorithm are selected to form the RFF feature. The RFF feature generation process is shown in Figure 4.

In this section, the dimensionality reduction processing of communication signals is divided into two steps: First, this paper chooses 11-dimensional features that contain 95% of the energy of the amplitude envelope of signal as the first step of dimensionality reduction processing. The energy proportion after dimensionality reduction of the original signal with PCA is shown in Table 1. Subsequently, the NCA method is used to select the features of the signal, and the 7-dimensional features with an NCA score greater than 1 are selected as the final RF fingerprint. This is the second step of the dimensionality reduction process.

The RFF training SVDD model of authentic devices was also used, and the training data included the communication signals (transient signals) of eight devices. In this paper, the SVDD model was established for eight devices under six SNRs (0 db, 5 dB, 10 dB, 15 dB, 20 dB, and 25 dB).

The following is a graphical example of how we can use the SVDD model to determine the legality of the device. Only a visualization of the SVDD model for device #2 is shown here. As shown in Figure 5, the x-axis represents the serial number of the sample points. Since this example is the signal of device #2, its serial number range is 31–60 (a total of 10 devices in the test set, and each device has 30 sample points). The y-axis represents the distance between the sample point and the center of the SVDD hypersphere, and the horizontal line in the figure is the decision threshold (i.e., the radius of the hypersphere). The radius of the hypersphere is determined by the WOA algorithm during model training. The discriminant threshold is determined by the supersphere radius.

It can be seen that under the condition of 5 dB, the distance between most positive samples and the center of the sphere is still below the decision threshold, but some of them fall above the decision threshold. Moreover, since the sample of device #4 is relatively close to the sample of device #2, the sample of device #4 is also relatively close to the decision threshold. It can be seen that at SNR = 10 dB, although a small number of positive samples are above the judgment threshold, the center of SVDD is far away from the sample points of other devices, which reduces the possibility of misjudgment in the model. When the SNR increases gradually, the performance of the SVDD model becomes better and better because the sample points are more concentrated in the feature space.

With these examples, we introduced how a single category of SVDD model works. However, as authentic devices often contain many categories, we need to build a multi-category device authentication model. The authentication process is shown in Figure 6. And the specific form of the RFF authentication model is shown in Figure 7. The authentication model is divided into three parts: RFF generator, K nearest neighbor finder, and SVDD discriminator. The RFF generator is used to convert transient signals into RFF features. The K-nearest neighbor finder is responsible for assigning the sample to be authenticated to the SVDD model of a specific device. It finds the category of the K samples closest to the sample to be certified in the RFF feature space and determines which SVDD model the sample to be certified is assigned to. The SVDD discriminator is composed of N SVDD models (N is the number of authentic devices), and each authentic device establishes an SVDD model. It provides a deny option to the system to deny illegal access to those rogue devices. The above three parts together constitute the authentication model. Its tasks include two aspects: 1. Denying illegal access by rogue devices; 2. Accurately identify authentic devices.

In order to ensure the reliability of the experimental results, and avoid experimental results being affected by a specific grouping of equipment, a total of five independent experiments were performed. Each group of experiments randomly selected two devices as rogue devices and the remaining eight devices as authentic devices. Finally, the average result of five independent experiments was taken as the experimental result. Through the experimental results, the authentication ability of this method to each authentic device and the detection ability of this method to each rogue device were analyzed. Table 2 shows the authentication success rate of eight authentic devices when using the SVDD authentication model.

Considering the completeness and conciseness of the experimental results, we only listed the maximum and minimum ASR in each group of experiments, as this can show both the best case of SVDD model for each group of data authentication and the worst case of each group of data authentication. This does not take up a great deal of space. When the SNR is lower than 5 dB, the difference between the best and worst results of each group of experiments is about 8%. With the increase of SNR, the distribution of features becomes more concentrated, and the gap between the best and worst results of each group of experiments becomes smaller and smaller. At a signal-to-noise ratio of 20 dB, the worst-case ASR was over 95%.

The comparison method selected in this paper is based on the sample average distance authentication model and the posterior probability SVM. The ASR curve of authentic devices is shown in Figure 8. We can see that the ASR of the three methods exceeds 40% when the SNR is greater than 1 dB. As the SNR increases, the performance of the three methods begins to improve significantly. During the 5–10 dB period, the ASR of the method of this paper achieved a rapid growth. This is because, in this interval, the feature distribution changes rapidly, and the characteristics of the similar sample points are accelerated. The SVDD model also performs better for training samples in the sample point set. It can be seen that when the SNR exceeds 15 dB, the ASR of this method exceeds 90%, which further proves the effectiveness of the proposed method.

In the authentication phase, we not only completed the correct authentication of the authentic devices, but also the detection and judgment of the rogue devices. We use rogue devices to attempt to log in to the authentication system, and then the authentication system gives the authentication results. In the specific operation, we will judge according to the distance of the SVDD model output in the authentication system. We use the hypersphere radius as a threshold, and for devices below the threshold, the system terminates its authentication operation. Table 3 shows the detection success rate of two rogue devices when the SVDD authentication model is used. It can be seen that the detection capability of the SVDD model for rogue devices is basically the same, with an average difference of 1–2% for five SNR.

The effect of the authentication system successfully detecting rogue devices is shown in Figure 9. We can see that our method is usually ten percent ahead of the comparison method. The performance difference is especially obvious in the case of low SNR. With the improvement of SNR, the gap starts to narrow. This is because the distribution of RFF characteristics of different categories becomes sparse in the case of high SNR. When the SNR is 15 dB, the RSR of all three methods reaches 90%. It is not difficult to find that the method in this paper has better authentication performance and detection performance than the comparison method, which is largely due to the advantage of the SVDD algorithm in data description—making full use of the sample tag information.

## 5. Conclusions

This paper introduces the theoretical basis and experimental demonstration of an RFF authentication model based on the SVDD algorithm. First, the NCA feature selection algorithm and SVDD algorithm are introduced. Their theory and application method are introduced in detail. Then we apply these two algorithms to the RFF authentication model. We have outlined the operation of the authentication model. The model is evaluated with real devices. Experimental results show that the RFF authentication model is reliable. From the above simulation experiments, it can be seen that the application of the RFF authentication model to the identity authentication of the MIoT device can achieve better performance, and due to its work in the physical layer, the security is also greatly guaranteed. In other words, the RFF authentication model can well meet the security performance requirements of a device in the MIoT.

Due to the time and space limitations, this paper only discusses the recognition method based on specific channel conditions. In the future, we are trying to use a functional model to describe the unique physical layer differences of the devices. Furthermore, the channel influence is separated from the function model, so that this technology can be used in a scenario where the channels are dynamic.

## Figures and Tables

**Figure 1 sensors-20-01213-f001:**
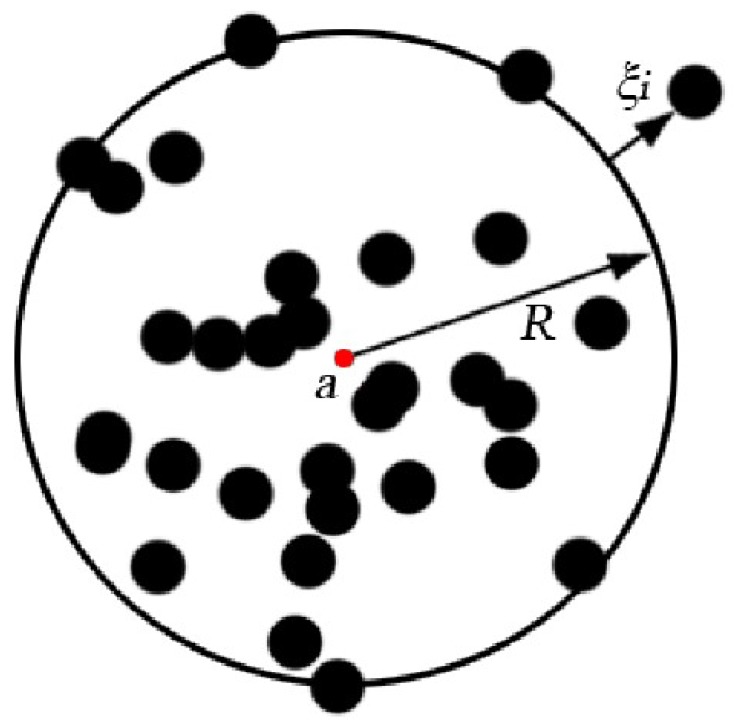
A support vector data description geometric model.

**Figure 2 sensors-20-01213-f002:**
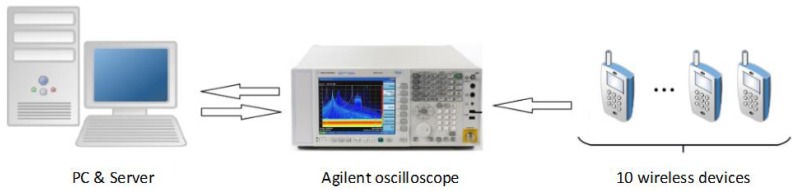
The experimental setup.

**Figure 3 sensors-20-01213-f003:**
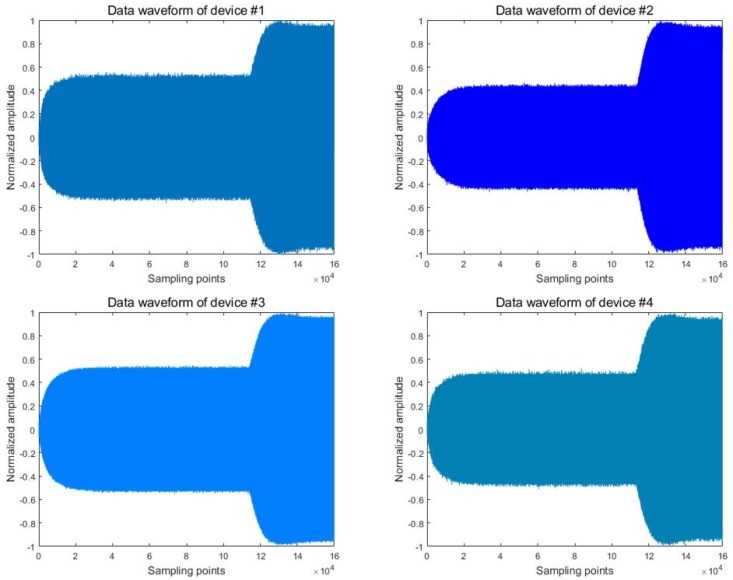
Transient signal waveforms of four wireless devices.

**Figure 4 sensors-20-01213-f004:**
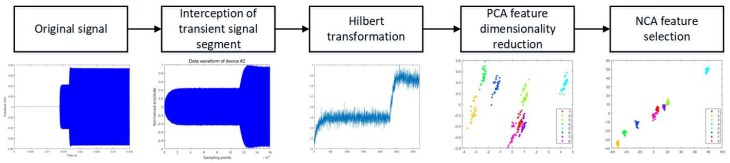
The radio frequency fingerprint (RFF) feature generation process. Neighborhood component analysis (NCA).

**Figure 5 sensors-20-01213-f005:**
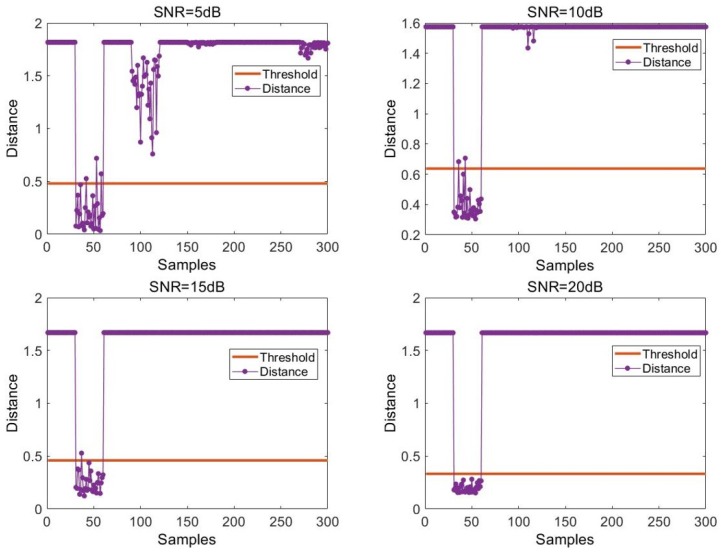
Schematic diagram of the distance between the sample point and the hypersphere. Signal-to-noise ratio (SNR).

**Figure 6 sensors-20-01213-f006:**
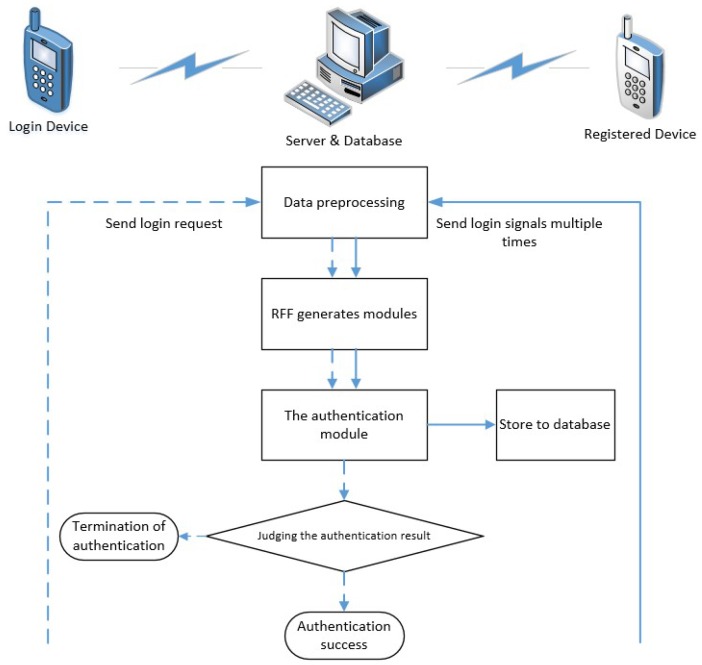
The process of the RFF authentication.

**Figure 7 sensors-20-01213-f007:**
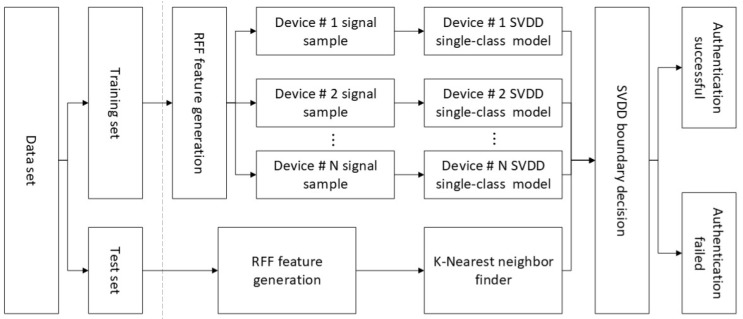
The specific form of the RFF authentication model. Support vector data description (SVDD).

**Figure 8 sensors-20-01213-f008:**
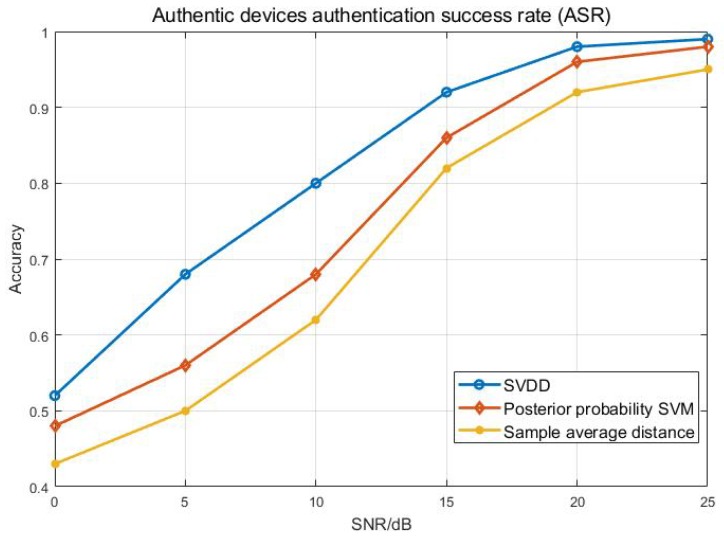
Authentic devices authentication success rate.

**Figure 9 sensors-20-01213-f009:**
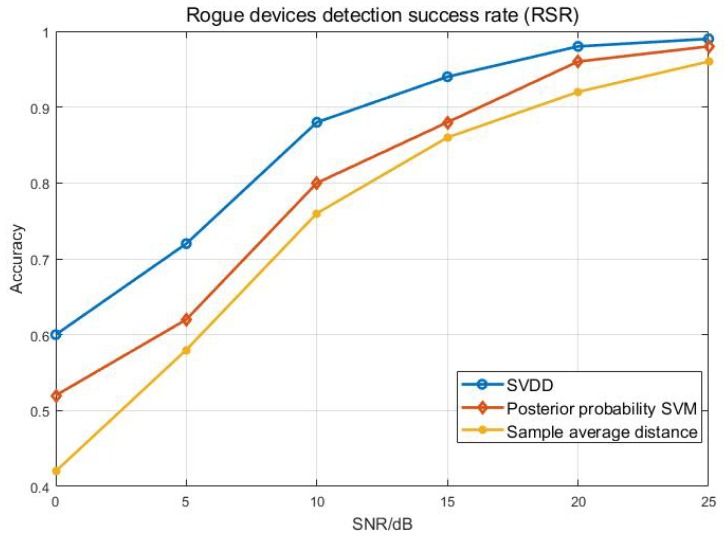
Rogue device detection success rate.

**Table 1 sensors-20-01213-t001:** Energy proportion of the feature dimensions after principal component analysis (PCA) dimensionality reduction.

Energy Proportion	85%	90%	95%
Feature dimension	6	8	12

**Table 2 sensors-20-01213-t002:** Authentic devices authentication success rate (ASR) for the SVDD model.

Experiment	${\mathrm{ASR}}_{\min}$ and ${\mathrm{ASR}}_{\max}$	SNR					
		0 dB	5 dB	10 dB	15 dB	20 dB	25 dB
1	$ASR_{\min}$	0.48	0.63	0.77	0.88	0.95	0.99
	$ASR_{\max}$	0.55	0.69	0.85	0.95	1.00	1.00
2	$ASR_{\min}$	0.41	0.61	0.78	0.86	0.94	0.98
	$ASR_{\max}$	0.49	0.7	0.84	0.92	0.99	1.00
3	$ASR_{\min}$	0.5	0.66	0.78	0.9	0.95	0.99
	$ASR_{\max}$	0.6	0.71	0.82	0.94	0.98	1.00
4	$ASR_{\min}$	0.48	0.67	0.79	0.92	0.97	1
	$ASR_{\max}$	0.51	0.71	0.83	0.96	1.00	1.00
5	$ASR_{\min}$	0.48	0.65	0.74	0.88	0.96	0.99
	$ASR_{\max}$	0.56	0.72	0.82	0.94	1.00	1.00

**Table 3 sensors-20-01213-t003:** Rogue devices detection success rate (RSR) for the SVDD model.

Experiment	Rogue Devices	SNR					
		0 dB	5 dB	10 dB	15 dB	20 dB	25 dB
1	#9	0.58	0.72	0.89	0.93	0.97	0.99
	#10	0.59	0.75	0.90	0.96	0.99	1.00
2	#2	0.64	0.76	0.92	0.95	0.98	0.99
	#6	0.61	0.70	0.86	0.91	0.96	0.99
3	#7	0.58	0.70	0.87	0.92	0.97	0.99
	#8	0.56	0.67	0.89	0.94	0.96	1.00
4	#1	0.65	0.76	0.92	0.96	0.98	0.99
	#4	0.58	0.67	0.87	0.91	0.97	1.00
5	#3	0.61	0.73	0.90	0.95	0.98	1.00
	#5	0.60	0.69	0.86	0.92	0.97	0.99

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
