# Peer review of "An Identity Authentication Method of a MIoT Device Based on Radio Frequency (RF) Fingerprint Technology"

_sensors, 2020, doi:10.3390/s20041213_

Round 1

Reviewer 1 Report

Although you state in your response that you have "made changes to this paper to clear the way for readers", the paper still has numerous grammatical errors that make reading of the article very difficult. There are numerous sentences that do not convey a complete thought. For example, (i) "However, since authentic devices often contain many categories, we need to build an RFF authentication model to authenticate authentic devices. The authentication process is shown in Figure 6." and (ii) "Introduced the experimental device and experimental procedures, followed by the performance indicators of the authentication system." Also, the use of the word "sample" seems to change meaning throughout the paper, which also provides a point of confusion.

You failed to add the model information of the radios to the paper.

In the authors' response, it states, "In this paper, a total of 5 independent experiments were performed. Each group of experiments randomly selected 2 devices as rogue devices and the remaining 8 devices as authentic devices. Finally, the average result of 5 independent experiments was taken as the experimental result." So, the results are presented as an average of trials. Thus, the authors have not addressed my initial concerns from my first review in which I stated the issue(s) with using of average/mean.The authors need to provide the individual results for each of the five experiments that used a different set of rogue radios. Also, these results should be further broken down by KNN mapping of a given rogue. For example, using rogue #9, the KNN classifier may or may not identify it as any one of the authentic devices at a given SNR; thus, the results should reflect this where applicable and not simply `average' across that and experiments that did not even use #9 as a rogue device.

The authors still do not provide a clear progression through their developed methodology and support it with figures. This greatly diminishes the ability of reader to: (i) follow along and clearly understand what is exactly being done, and (ii) repeat/implement the developed methodology. A couple examples that support my position: (i) the use of PCA without specifying the number of components used nor what the specific input is into the PCA and latter NCA selection processes are.(ii) The authors state, "the sample of device #4 is relatively close to the sample of device #2;" however, this is not made clear to the reader nor supported using evidence/results.

Author Response

Thank you for your comments on this paper. My reply to you can be seen in the.pdf file.

Reviewer 2 Report

The resubmitted manuscript is well modied, and the reviewer has no further comments.

Author Response

Thank you for your comments on this paper. 

Reviewer 3 Report

The paper is about an interesting topic, however, it must be improved in order to be presented in a journal.

1 - The introduction must clearly describe the purspose of the paper, and identify why it should be published.

2 - Section 2 is very confusing an discuss different subjects that are not straight related to the paper. I suggest to reduce it and focus on the object of the paper.

3 - More details about the RF problem is required.

4 - Authors must justify their work in a real scenario, since I could not identify how this "technology" could work in a scenario where channels are dynamic.

5 -  The methodology as well as the results/discussion must be improved.

Author Response

(The authors gave the same response as above.)

Reviewer 4 Report

This paper applies RF fingerprint technology to MIoT devices’ identity authentication. On the one hand, a RF fingerprint authentication model is proposed in this paper, and the actual data are collected to test the effectiveness and practicality of the algorithm. The whole paper is clear and coherent. On the other hand, it has a rich demonstration of experimental results. Generally, this paper is well organized and easy to follow. Although this paper contains some deficiencies, however, the author presented his approach clearly, it can be published if the authors make suitable revision. Following are some suggestions for improving the paper:

1) First, in the third section of the paper, the paper mentioned the whale swarm optimization algorithm, but did not introduce the specific details of the whale swarm optimization algorithm. I hope the author can give a specific calculation process.

2) The SVDD method is used in this paper, but the development of SVDD is not fully introduced in the second section of the paper. I hope the author could complete the development of the SVDD algorithm.

3) In the simulation experiment part, this paper uses the principal component analysis method to reduce the feature dimension. But I am not sure what standard the authors use to determine the feature dimension after dimensionality reduction. Please explain this.

4) In Figure 4, the authors mention intercepting transient signals from the original signal. How does this paper intercept the original signal?

5) In this paper, the SVDD model is used for RF fingerprint authentication. According to formula (8), we know that SVDD uses the kernel function method. But the authors have not yet specified what kind of kernel function is used in this paper?

6) There are some language mistakes and misprints – the manuscript should be checked carefully by native English speaker.

Author Response

(The authors gave the same response as above.)

Round 2

Reviewer 1 Report

This is the third or fourth time I have reviewed this paper and I have to say that the topic is of interest and I do think the contribution is there; however, the quality of the paper is poor. As with all of my reviews, my position still remains that the use of the English language is very poor in this paper. The are numerous points within the paper that a sentence does not make sense, because of poor grammar and lack of understanding of the English language. Again, as before, this puts an unnecessary burden on the reader and distracts from the technical contribution of the work. Some examples of the poor use of the English language include: proper pluralization of words, improper or missing punctuation, lack of a complete thought within a sentence, and inclusion of the word 'the' where necessary. There also remains numerous typos such as: lack of space at the end of a sentence, lack of consistency in capitalization of words, e.g., Internet of things, internet of things, and Internet of Things. 

Please clarify what is meant by the following on line 38: The technology is called RF fingerprint authentication. What 'technology' are you referring to? 

From line 37: What is meant by 'difference'? 

Lines 26 through 30 provide redundant statements. 

What is the meant by, In literature, authors identify the amplitude of transient signals use Bluetooth and IEEE 802.11 transceiver devices to. This is a great example of poor grammar and use of the English language. 

The sentence on line 139, beginning with 'Secondly' does not make sense.

Define 'LOO'.

In section 3.1, the use of 'A' to refer to a matrix is confused with the use of 'a' as in 'A car is in the driveway.'

How are researchers enlightened? Please explain the sentence that ends on line 176.

You still fail to provide the specific model information for the Motorola radios used despite being asked to provide this information within the paper.

The explanation of the WOA starting on line 228 through the sentence that ends on line 231 is very poor. Again, poor use of the English language. 

The following sentence is also poorly written and makes no sense in English, 'Otherwise, it is the identity illegal rogue device.'

Provide an explanation of 'D' from equation 10 versus its use in equations 15 and 17. Are they equal?

The font in Figure 2 is too small and is unreadable.

Explain what is meant by, 'The experimental device is shown in Figure 2.' Do you mean, 'The experimental setup is shown in Figure 2.'?

Sentences starting on line 293 and ending on line 295 are redundant. 

For the PCA dimensional reduction, are you computing the principle components of the amplitude envelope? If so, what is the original dimensionality of this envelope?

When referring to the 'original signal' on line 326, are you referring to the collected signal, its transient, or the amplitude envelope?

Explanation of how Figure 7 fits into or relates to Figure 6 is needed.

There are missing works within the area of RF fingerprinting authentication that need to be cited and addressed within this paper.

Author Response

Thank you for your comments on this paper. Our response is in the word file.

Reviewer 3 Report

The authors improved the paper as recomended. However, the weak point of the paper persists:  how this
"technology" could work in a scenario where channels are dynamic.

Author Response

(The authors gave the same response as above.)

Reviewer 4 Report

Authors have successfully addressed my concerns.

Author Response

Thank you for your comments on this paper.

This manuscript is a resubmission of an earlier submission. The following is a list of the peer review reports and author responses from that submission.

Round 1

Reviewer 1 Report

The submitted manuscript titled, "Identity Authentication Method of MIoT Device Based on RF Fingerprint Technology" proposes a transmitter identity authentication approach using RF fingerprints and an SVDD classifier coupled with NCA-based feature selection. The manuscript results are generated using eight (8) 'authentic' and two (2) 'rogue' devices. 

Author Response

Response to Reviewer 1’s Comments

1. There are grammatical and other English language related errors that are too numerous to highlight here. These errors distract from the paper and its contribution; thus, the paper needs to be re-written. Acronyms are not introduced with the accompanying phrase they are to represent. For example, `MIoT', which represents `mobile Internet of Things' or `RFF', which does not appear to be defined anywhere in the paper. Acronyms are also misspelled or have typos. For example,`MioT' versus MIoT.

I am very sorry that some grammatical errors have affected your reading due to the negligence of the authors. We have made some changes to this paper to clear the way for readers. And the English abbreviations have been checked, so that each abbreviation has its corresponding English full name.

2. Upon reviewing the list of cited works (i.e., references), there seems to be an oversight in terms of papers that present results and/or methodologies for RF fingerprint-based authentication. This includes papers that have been published over the past two years. For example, the paper titled, `Analysis of Classification Methods Based on Radio Frequency Fingerprint for Zigbee Devices' and `Design of a Hybrid RF Fingerprint Extraction and Device Classification Scheme'. It is very important that the authors' literature review capture published works within the area of RF fingerprinting and device authentication.

Thank you for recommending excellent papers in the field of RFF to us recently. Through reading these papers, we have also revised and supplemented Section 2 to ensure the Related work is complete and comprehensive.

3. In terms of the presented results, the authors claim that a 90% success rate is achieved in authenticating the identities of the authentic and rogue devices at SNR=15 dB. On the surfacethese results appear encouraging, but are misleading upon further examination. The authors' present the average authentication rates and not the authentication rates for the individual devices(authentic nor rogue). The average provides the `central' trend of the results, which makes it sensitive to extreme values. Thus, the average results could be obscuring poor performing authentication performance for an individual device, whether authentic or rogue. The authors must present per device authentication performance; otherwise, the true contribution of the presented work cannot be appreciated nor analyzed.

We are deeply inspired by your questions. Therefore, we added the simulation results (ASR and RSR) of SVDD model for each device and discussed them in Section 4. And use this result to observe the RFF authentication model's ability to authenticate each authentic device and the ability to detect each rogue device.

Reviewer 2 Report

This work introduces a RF fingerprint based authentication method for IoT devices. The idea is interesting. I have following comments:

The contribution of this work should be highlighted, especially compared to recent research on RF fingerprint authentication. I can hardly find the mobility of mobile Internet of Things (MIoT) in this work. Which kind of IoT device are used for experiment? Does the dataset consist the received MIoT signals at different locations or in movement? It seems that in different SNR scenarios, the threshold changes. How to get the threshold in Figure 3? It is not clear to find the description of obtaining authentication threshold in Section 3 (Method). Section 2 (Related work) should be reorganized.

Author Response

Response to Reviewer 2’s Comments

This work introduces a RF fingerprint based authentication method for IoT devices. The idea is interesting. I have following comments:

(1) The contribution of this work should be highlighted, especially compared to recent research on RF fingerprint authentication.

(2) I can hardly find the mobility of mobile Internet of Things (MIoT) in this work. Which kind of IoT device are used for experiment? Does the dataset consist the received MIoT signals at different locations or in movement?

(3) It seems that in different SNR scenarios, the threshold changes. How to get the threshold in Figure 3? It is not clear to find the description of obtaining authentication threshold in Section 3 (Method).

(4) Section 2 (Related work) should be reorganized.

Thank you for your Suggestions. We have answered your questions about this paper:

(1) Since (1) and (4) are some comments on Related work, we have improved Section 2. Based on the original, we have included representative papers on radio frequency fingerprint authentication since 2012. From the perspective of methods and analysis of experimental results, the development of RF fingerprint authentication in recent years is introduced. At the same time, we also introduce the differences and improvements of the research in this paper from previous RF fingerprint authentication methods.

(2) In the experiment, the experimental equipment we used was a certain model of Motorola mobile phone. Use a cable to connect them to the oscilloscope. To restore the real scene, we add white Gaussian noise during the simulation.

(3) Here we also supplement the description of the method in this paper in Section 3. For the parameter selection of the SVDD model, we use the WOA algorithm that has been popular in recent years. The hypersphere parameters of SVDD determine the decision threshold of the model, and the hypersphere parameters are determined based on the training samples.

(4) This comment to this recommendation is detailed in response (1).

Round 2

Reviewer 1 Report

Please see the attached document containing my comments.

Reviewer 2 Report

Although authors have modified the paper concerning my comments, I still have questions about their experiment:

1. Which kind of MOTOROLA mobile phone signal is used for RF identifications? (GSM? 4G-LTE?)

2. Authors use cables to connect the oscilloscope to the mobile phones. The obtained results can hardly be verified in real environment with wireless transmissions.
Authors should study state-of-art about RF identifications. Very few works directly use cable connection for signal collections and RF identifications. This is completely NOT mobile IoT! Authors should carry out experiments in real wireless and mobile environments.